# WeChat-assisted strategies for personalized health management in patients with AECOPD: A randomized controlled trial

**Lili Zhou, Cailing Song, Wenpeng Xu, Ruirui Wang, Wei Zhang** [ORCID]*

No.2 People's Hospital of Fuyang City, Fuyang Infectious Disease Clinical College of Anhui Medical University, Fuyang, Anhui, China

* 94499420@qq.com

## Abstract

### Objective

To investigate the application effect of personalized health management strategies based on the WeChat platform in improving the overall health status of patients with Acute Exacerbation of Chronic Obstructive Pulmonary Disease (AECOPD), aiming to provide scientific evidence for enhancing the health management of AECOPD patients.

### Methods

From February 2024 to September 2024, a total of 120 AECOPD patients treated in the Comprehensive Ward of Respiratory and Critical Care Medicine at the Second People's Hospital of Fuyang City were selected. They were divided into an observation group and a control group, each consisting of 60 patients, using the random number table method. The control group received conventional care, while the observation group additionally utilized the WeChat platform for personalized health education, symptom monitoring guidance, patient-doctor interaction, and other interventions. Post-intervention comparisons were compared between the two groups regarding self-management ability scores, quality of life scores, readmission rates, and emergency visit frequencies.

### Results

After the intervention, the self-management ability score of the observation group was significantly higher than that of the control group, with higher scores in symptom management, daily life management, emotional management, information management, and self-efficacy. Quality of life was also better in the observation group compared to the control group, particularly in physiological, psychological, sociocultural,

**Data availability statement:** All relevant data are within the manuscript and its Supporting information files.

**Funding:** This work was supported by the Scientific Research Project of Fuyang Municipal Health Commission (Grant Number: FY2023-102). The funders had no role in study design, data collection and analysis, decision to publish, or preparation of the manuscript. There was no additional external funding received for this study.

**Competing interests:** The authors have declared that no competing interests exist.

and environmental aspects. The readmission rate was 8.33% (5/60) and emergency visits were 6.67% (4/60) in the observation group, both significantly lower than the control group's 21.67% (13/60) and 20.00% (12/60). Clinical outcomes showed 13.3% absolute risk reductions in both readmissions and emergency visits (NNT = 7.5 for both). Nursing satisfaction in the observation group reached 93.33%, significantly higher than 76.67% in the control group, with all differences being statistically significant (P < 0.05).

## Conclusion

The auxiliary management model based on the WeChat platform effectively enhances the self-management capabilities of AECOPD patients, improves their quality of life, reduces hospital readmissions and emergency visits due to disease exacerbations, increases nursing satisfaction, optimizes medical resource allocation, and promotes long-term health management among patients, thus possessing substantial clinical promotion value.

## Introduction

Chronic Obstructive Pulmonary Disease (COPD) is a common chronic condition characterized by persistent respiratory symptoms and airflow limitation [1,2]. Its acute exacerbation phase (AECOPD) severely impacts patients' quality of life and increases healthcare burdens [3]. Globally, COPD is the third leading cause of death, responsible for 3.23 million deaths and 74.4 million DALYs in 2019 [4]. AECOPD accounts for 60–70% of COPD-related expenditures and drives most hospitalizations [5].

In China, the 2021 Global Burden of Disease Study reported 50.59 million COPD cases and 1.29 million deaths—nearly 25% of global cases, rural areas face higher mortality (66.24 vs 47.30 per 100,000 urban) [4]. Each severe AECOPD episode costs $1,400–2,100 with 7–10 extra hospital days, and 22% of patients re-exacerbate within 30 days [6].

Effective self-management reduces exacerbations and hospitalizations [7], yet current models struggle with personalization and real-time monitoring, traditional methods (e.g., face-to-face or telephone follow-ups) lack continuity [8]. WeChat's immediacy and interactivity (1.3 billion active users) enable innovative solutions [9,10], such as personalized education and symptom monitoring [11], proven to enhance self-management in chronic diseases [12–14].

This study evaluates a WeChat-based program for AECOPD patients with four key innovations: (1) 6-month intervention assessing sustainability; (2) age/gender-stratified randomization; (3) multi-dimensional interactions (daily logs, biweekly videos, monthly seminars); (4) emergency call functionality. These address fragmentation in traditional care.

## Materials and methods

### General information

The sample size was calculated a priori using G*Power 3.1.9.7 (Heinrich-Heine-Universität Düsseldorf, Germany). Based on previous RCT data [15,16] showing a 20% between-group difference in COPD Self-Management Scale scores (control: $50 \pm 12.5$ vs intervention: $60 \pm 12.5$), a two-tailed independent t-test with $\alpha = 0.05$ and 80% power ($\beta = 0.20$) indicated 49 participants per group. To account for potential 10% attrition, we enrolled 60 patients per arm (total $N = 120$), exceeding the minimum requirement. A total of 120 AECOPD patients treated in the Comprehensive Ward of Respiratory and Critical Care Medicine at the Second People's Hospital of Fuyang City between February 2024 and September 2024 were selected for this study (Fig 1). The study received comprehensive review and formal approval from the Ethics Committee of the Second People's Hospital of Fuyang City (Approval Number: 20231112034), which functions as the Institutional Review Board (IRB) equivalent and is fully compliant with international ethical standards including ICH-GCP guidelines. The committee maintains institutional independence with multidisciplinary membership (including clinicians, researchers, legal experts, and community representatives), and follows standardized procedures for protocol review, risk assessment, and ongoing monitoring. All committee members have completed certified ethics training, and no conflicts of interest were declared during the review process.All research procedures strictly adhered to the relevant provisions of the Helsinki Declaration. All participants provided written informed consent. Patient data were handled according to ethical standards to ensure confidentiality and anonymity. No personal identifying information was used during analysis or reporting. For participants with limited literacy, the consent form was read aloud by a researcher, and a witness (independent healthcare staff) confirmed their voluntary agreement via signature. The ethics committee waived the need for parental/guardian consent as all participants were adults (aged 38–75 years).

Inclusion criteria were: (1) meeting the diagnostic criteria for AECOPD as outlined in the "Chinese Guidelines for Diagnosis and Management of Chronic Obstructive Pulmonary Disease in Primary Care (2024 Edition)" [17]; (2) aged between 38 and 75 years; (3) possessing basic skills for using smartphones and WeChat applications; (4) voluntarily signing an informed consent form. Exclusion criteria included: (1) having other severe systemic diseases such as heart, liver, or kidney disorders; (2) suffering from mental disorders or cognitive impairments that affect self-management capabilities; (3) being unable to complete the entire research process.

Patients were randomly assigned 1:1 to observation or control groups ($n = 60$ each) via a computer-generated random number table, with stratified randomization by gender (male/female) and age ($\leq 60 / > 60$ years). To control for potential residual confounding, baseline comparisons utilized ANCOVA (continuous variables, adjusted for stratification factors) and multinomial logistic regression (categorical variables). As shown in Table 1, no significant differences existed between groups in demographics (age, gender), clinical characteristics (disease duration, comorbidities), or smoking history (all $P > 0.05$), confirming successful randomization.Due to the nature of behavioral interventions (e.g., WeChat platform usage cannot be concealed from participants), blinding of patients and caregivers was not implemented. However, outcome assessors were blinded to group allocation during data analysis to ensure objectivity. While this design may introduce expectation bias, the use of standardized tools (e.g., COPD Self-Management Scale) and objective outcomes (e.g., readmission rates) partially mitigates this limitation.

### Nursing methods

**Control group.** Patients in the control group received standard multidisciplinary care for AECOPD, including: (1) Medication management: Bronchodilators (nebulized salbutamol/ipratropium, 4 times/day), oral corticosteroids (prednisone 30 mg/day for 7–10 days), and antibiotics when indicated (per GOLD guidelines); (2) Structured health education: A 30-minute face-to-face session at discharge covering inhaler technique, smoking cessation, and exacerbation recognition; (3) Routine follow-up: Telephone consultation at 2 weeks and scheduled outpatient visit at 1 month post-discharge; (4) Basic monitoring: Paper-based symptom diary (no digital tracking) and provision of emergency contact numbers.

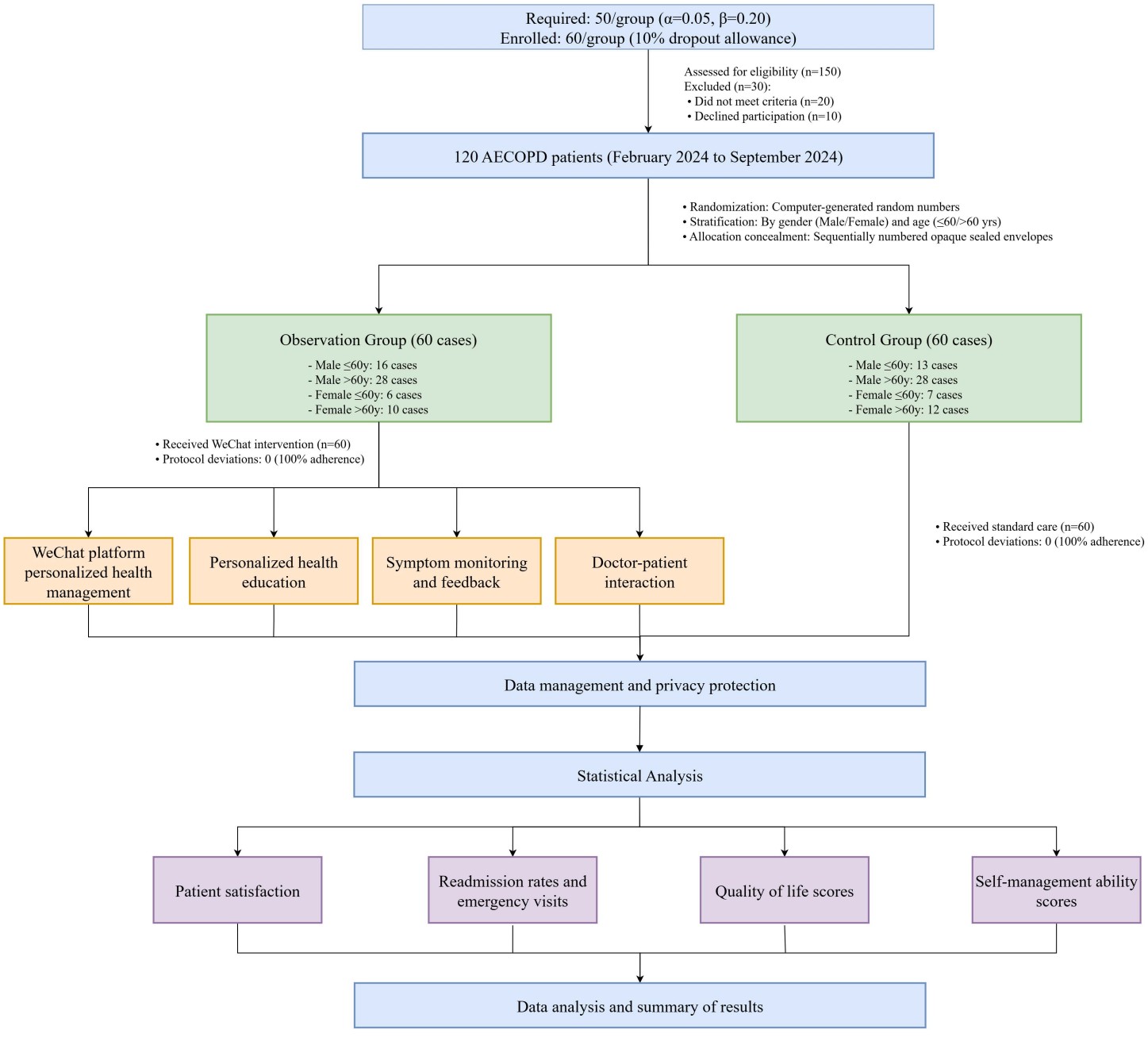

**Fig 1. Flowchart.**

**Observation group.** On top of the conventional nursing care, the observation group integrated comprehensive intervention management via the WeChat platform to ensure that each patient received the full six-month intervention. Specific measures were as follows:

(1) Establishment and Functions of the WeChat Platform. ① A member from the medical team who is proficient in WeChat operations and possesses excellent communication skills was appointed as the dedicated WeChat administrator. This

**Table 1. Baseline characteristics comparison.**

| Characteristic | Observation Group (n=60) | Control Group (n=60) | *P* value |
|---|---|---|---|
| **Demographics** | | | |
| Age(years,mean±SD) | 63.77±6.75 | 64.92±7.96 | 0.412* |
| Gender[n(%)] | | | 0.538# |
| Male | 44(73.33) | 41(68.33) | |
| Female | 16(26.67) | 19(31.67) | |
| **Clinical Variables** | | | |
| Duration of Disease(years,mean±SD) | 5.12±2.32 | 4.95±2.11 | 0.682* |
| Comorbidities | | | 0.621# |
| Hypertension | 22 (36.67) | 25 (41.67) | |
| Diabetes | 15 (25.00) | 12 (20.00) | |
| Smoking history | | | 0.832# |
| Current smoker | 18 (30.00) | 20 (33.33) | |
| Ex-smoker | 25 (41.67) | 23 (38.33) | |
| Never smoker | 17 (28.33) | 17 (28.33) | |

Note:

*ANCOVA adjusted for age, gender, and disease duration; # Multinomial logistic regression adjusted for age and gender

individual received specialized training to daily manage and maintain the WeChat group, including responsibilities such as disseminating information, answering questions, and recording data. ② Using a departmental public mobile phone, a WeChat group named "AECOPD Health Management" was created. Members of this group include doctors, nurses, patients, their families, and research team members. Each patient and their family members were ensured to join the group, and they were guided on how to use WeChat for communication and data submission. ③ The administrator posted health tips daily before 9 AM in the group, such as medication reminders and breathing exercise instructions. Every Tuesday and Thursday, articles or videos about the basics of AECOPD, the latest research developments, and preventive measures are shared with the group. Patients could ask questions anytime in the group, with healthcare professionals responding within 24 hours to ensure information accuracy and timeliness.

(2) Personalized Health Education. ① Based on each patient's condition, age, lifestyle, and other factors, doctors and nurses jointly develop personalized health education plans. These plans are sent to patients via WeChat official accounts or mini-programs and are regularly updated and adjusted. ② An online health seminar is organized once a month, where experts are invited to discuss AECOPD prevention and treatment knowledge, with a Q&A session to encourage patient questions and participation in discussions. Seminar notifications are posted in the WeChat group one week in advance. Patient feedback was collected post-seminar to iteratively improve content.. ③ Patients were asked to record their daily symptom changes (such as cough frequency, sputum color, dyspnea severity), medication usage, and lifestyle adjustments (such as exercise levels, dietary habits, etc.). Patients must upload their health diaries to the WeChat group before 8 PM, where they are reviewed by responsible nurses who provide feedback.

(3) Symptom Monitoring and Feedback. ① Each patient is provided with an electronic symptom log, which includes key indicators such as cough frequency, sputum color, and degree of breathlessness. Patients are required to fill out this log daily and upload it to the WeChat group before 8 PM, allowing healthcare professionals to monitor symptoms in real-time. ② Healthcare professionals analyze the symptom data reported by patients. If any abnormalities (such as worsening symptoms) are detected, they immediately contact the patient via WeChat to provide targeted advice or adjust the treatment plan, documenting the process accordingly. In cases where patients exhibit acute exacerbation

symptoms, they can use a one-click emergency alert feature on WeChat to call for immediate medical assistance and simultaneously notify the attending physician to ensure timely treatment.

(4) Patient-Doctor Interaction. ① Each patient is assigned a responsible nurse who manages daily communication and care. Patients can consult their responsible nurse about their condition via WeChat at any time to receive personalized health guidance. ② A telephone or video follow-up is scheduled every two weeks to monitor the patient's recovery progress, answer questions, and adjust health plans as necessary. After each follow-up, the patient's feedback is recorded, and continuous tracking of their health status is conducted. ③ A satisfaction survey is conducted monthly to gather patients' opinions and suggestions regarding the service content. Management models are continuously optimized based on survey results to improve service quality.

(5) Data Management and Privacy Protection. All data transmitted through the WeChat platform is encrypted to ensure patient privacy and security. Posting permissions within the WeChat group are restricted to healthcare professionals and administrators to prevent interference from irrelevant individuals. The research team regularly backed up patient-submitted chat records and health data for analysis.

A concise timeline of the 6-month WeChat-based intervention is presented in Fig 2.

## Observation indicators

(1) Self-Management Ability Score: The COPD Self-Management Scale [18] evaluates five domains using a 5-point Likert scale (1 = complete absence to 5 = complete presence): symptom management (25% weight, e.g., "I record symptoms via WeChat diaries"), daily life management (25%), emotional management (20%, included based on Wu et al.'s evidence of psychological mediation effects), information management (15%, specifically health information application), and self-efficacy (15%). Total scores range from 51 to 255, with higher scores indicating better outcomes. The scale demonstrated reliability (α = 0.89) in Chinese COPD populations.

(2) Quality of Life Score: The General Comfort Questionnaire (GCQ), developed by Katharine Kolcaba based on her Comfort Theory [19], evaluates holistic comfort through physical, psycho-spiritual, environmental, and social dimensions. This 30-item instrument (physiological = 5 items, psychological = 10, sociocultural = 8, environmental = 7) uses a 4-point Likert scale (1 = "very unacceptable" to 4 = "fully acceptable"), with higher scores indicating better quality of life.

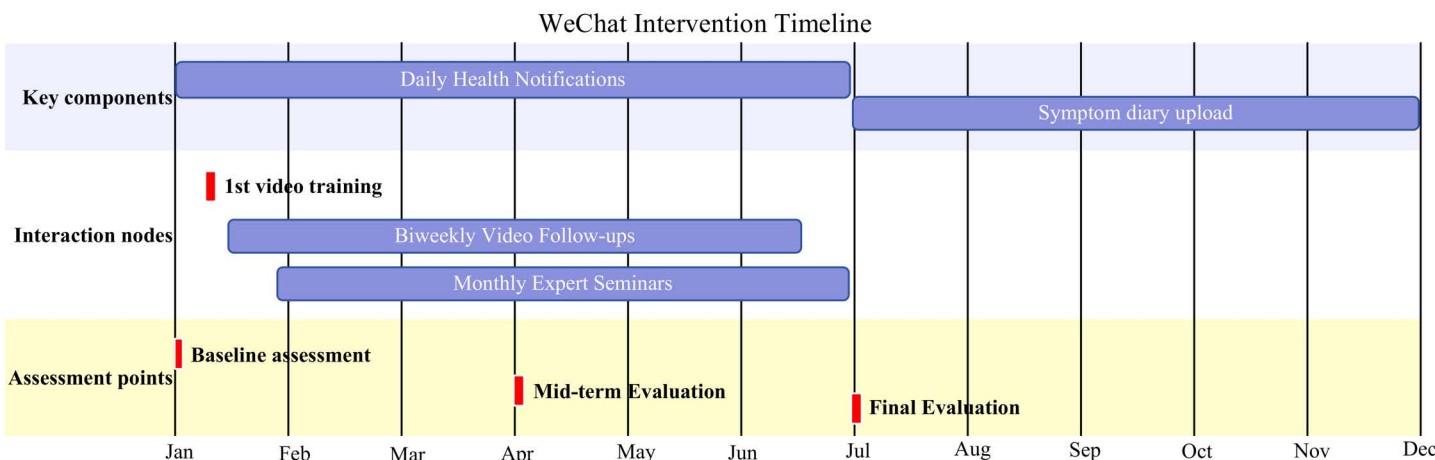

**Fig 2. WeChat intervention timeline.**

The scale has been validated internationally in chronic respiratory diseases (α = 0.89–0.93) [20] and specifically in Chinese COPD populations (α = 0.96) [18]. In our study, the GCQ demonstrated excellent reliability (α = 0.87), confirming its applicability for assessing WeChat-based interventions targeting comprehensive comfort improvement.

(3) Rehospitalization Rate and Emergency Department Visits: The rehospitalization rate and the frequency of emergency department visits due to acute exacerbations are statistically analyzed for both groups before and after the intervention.

(4) Patient Satisfaction: The Nursing Satisfaction Questionnaire, developed by our hospital, is used to assess patient satisfaction in both groups. The questionnaire has a total score of 100 points, categorized as follows: below 60 points as dissatisfied, 60–80 points as fairly average, 81–90 points as satisfied, and above 90 points as highly satisfied. The formula for calculating nursing satisfaction is: Nursing Satisfaction = (Number of Satisfied + Number of Highly Satisfied)/ Total Number of Cases * 100%. The Nursing Satisfaction Questionnaire was developed through expert consultation and patient feedback, demonstrating strong psychometric properties: Cronbach's α of 0.91 (total scale) and 0.83–0.88 (subscales), test-retest reliability (ICC = 0.89, 95% CI: 0.82–0.93), and construct validity confirmed by confirmatory factor analysis (CFI = 0.95, RMSEA = 0.06). Content validity was established through expert review (CVI > 0.8 for all items).

## Statistical analysis

This study adhered to the intention-to-treat (ITT) principle, with all randomized participants included in primary outcome analyses. Missing data were handled using multiple imputation. Baseline characteristics were compared using: ANCOVA for continuous variables (adjusted for age and gender stratification factors), and multinomial logistic regression for categorical variables (adjusted for age and gender). All models included disease duration as an additional covariate to control for residual confounding. To isolate the intervention effect from temporal trends, difference-in-differences (DID) analysis was performed using generalized estimating equations (GEE) with group × time interaction terms, adjusting for the same covariates. Full results of pre-post changes and DID estimates are provided in Supplementary S1 Table. Statistical analysis was performed using SPSS 26.0 software. For dichotomous outcomes (e.g., readmission rates), we calculated absolute risk reductions (ARR) with 95% confidence intervals and number needed to treat (NNT). For continuous data, between-group comparisons were expressed as mean differences (MD) with 95% CIs, while raw values are presented as mean ± standard deviation (SD), analyzed using the t-test. For categorical data, values are presented as [n(%)], and the chi-square ($\chi^2$) test was used for comparisons. A significance level of $P < 0.05$ was adopted.

To mitigate self-reporting biases in study design: (1) Questionnaire design: Incorporated 3 reverse-scored items and balanced positive/negative wording; (2) Objective anchors: Planned correlations with clinical indicators (FEV$_1$%, medication adherence logs); (3) Response monitoring: Predefined criteria for detecting extreme/random responses (e.g., > 80% identical scores); (4) Blinding: Outcome assessors were masked to group allocation during data collection.

Complete baseline raw data are available in Supplementary S3 Table.

## Results

### Comparative analysis of self-management ability scores between the two groups

After the intervention, the self-management ability score in the observation group was significantly higher than that in the control group. Scores for symptom management, daily life management, emotional management, information management, and self-efficacy were all higher in the observation group compared to the control group, with statistically significant differences ($P < 0.001$) (Table 2).Self-reported improvements were validated against objective measures: Self-management scores correlated with FEV$_1$% improvement (r = 0.39, 95% CI: 0.28–0.49), GCQ scores with medication adherence (r = 0.35, 0.24–0.45), Satisfaction scores with reduced emergency visits (r = −0.29, −0.39–0.18) (all P < 0.05, Supplementary S2 Table).

**Table 2. Comparison of self-management ability scores between groups (mean±SD).**

| Group | Symptom Management | Daily Life Management | Emotional Management | Information Management | Self-Efficacy | Total Score |
|---|---|---|---|---|---|---|
| Observation (n=60) | 33.57±5.12 | 43.57±6.07 | 41.80±5.79 | 29.43±5.06 | 30.94±5.19 | 183.56±10.25 |
| Control (n=60) | 28.75±5.11 | 39.86±6.14 | 37.79±5.46 | 25.36±4.21 | 27.41±4.76 | 165.82±11.78 |
| t | 5.161 | 7.191 | 7.363 | 4.790 | 3.883 | 8.800 |
| P | <0.001 | <0.001 | <0.001 | <0.001 | <0.001 | <0.001 |

## Comparative analysis of quality of life scores between the two groups

After the intervention, the quality of life scores in the observation group were higher than those in the control group. Specifically, scores for physiological, psychological, sociocultural, and environmental domains were all higher in the observation group compared to the control group (*P*<0.001) (Table 3).

## Comparative analysis of rehospitalization rates and emergency department visits between the two groups

The rehospitalization rate in the observation group was 8.33% (5/60) versus 21.67% (13/60) in controls, yielding an absolute risk reduction (ARR) of 13.34% (95% CI: 1.2% to 25.5%), with a number needed to treat (NNT) of 7.5. Similarly, emergency visit rates were 6.67% (4/60) vs. 20.00% (12/60), corresponding to an ARR of 13.33% (95% CI: 1.1% to 25.6%) and NNT=7.5. These risk differences were statistically significant ($\chi^2$=4.183, P=0.041 for rehospitalization; $\chi^2$=4.615, P=0.032 for emergency visits) (Table 4).

## Comparative analysis of nursing satisfaction between the two groups

The nursing satisfaction rate for patients in the observation group was 93.33% (56/60), which is significantly higher than the control group's rate of 76.67% (46/60). The difference was statistically significant ($\chi^2$=6.536, P=0.011) (Table 5).

**Table 3. Comparison of quality of life scores between groups (mean±SD).**

| Group | Physiological | Psychological | Sociocultural | Environmental | Total Score |
|---|---|---|---|---|---|
| Observation (n=60) | 14.78±3.67 | 31.51±6.10 | 22.53±4.12 | 20.87±3.66 | 92.87±7.65 |
| Control (n=60) | 12.32±2.92 | 27.26±5.86 | 18.27±3.88 | 16.19±2.89 | 80.62±6.39 |
| t | 4.063 | 3.892 | 5.831 | 7.774 | 9.520 |
| P | <0.001 | <0.001 | <0.001 | <0.001 | <0.001 |

**Table 4. Comparative analysis of rehospitalization rates and emergency department visits between the two groups.**

| Group | Rehospitalization rate | Emergency department visit rate |
|---|---|---|
| Observation[n(%)] | 5 (8.33) | 4 (6.67) |
| Control[n(%)] | 13 (21.67) | 12 (20.00) |
| Absolute Risk Reduction (ARR, %) | 13.34 (1.2 - 25.5) | 13.33 (1.1 - 25.6) |
| Number Needed to Treat (NNT) | 7.5 | 7.5 |
| $\chi^2$ value | 4.183 | 4.615 |
| P value | 0.041 | 0.032 |

NOTE: ARR: Absolute Risk Reduction; NNT: Number Needed to Treat; 95% confidence intervals in parentheses.

**Table 5. Comparison of patient satisfaction between the two groups[n(%)].**

| Group | Very Satisfied | Satisfied | Average | Dissatisfied | Overall Satisfaction |
|---|---|---|---|---|---|
| Observation (n = 60) | 30 (50.00) | 26 (43.33) | 3 (5.00) | 1 (1.67) | 56 (93.33) |
| Control (n = 60) | 20 (33.33) | 26 (43.33) | 10 (16.67) | 4 (6.67) | 46 (76.67) |
| χ² Value | – | – | – | – | 6.536 |
| P Value | – | – | – | – | 0.011 |

## Discussion

AECOPD is a significant cause of high morbidity and mortality worldwide [21–23]. Effective self-management is crucial for reducing acute exacerbations, improving quality of life, and lowering healthcare costs. Traditional care models often fail to meet patients' personalized needs, particularly in terms of long-term management and real-time monitoring. In recent years, with the development of mobile internet technology, social platforms such as WeChat have gradually become new tools for health management due to their convenience and immediacy.

This study found that the self-management ability scores of patients in the observation group were significantly higher than those in the control group, as evidenced by improvements in symptom management, daily life management, emotional management, information management, and self-efficacy. This indicates that the WeChat platform not only provides personalized health education content but also enhances patient engagement and adherence through interactive communication [24]. Regularly pushed disease knowledge, medication guidance, and rehabilitation training videos within the WeChat group helped patients better understand their conditions and acquire proper self-management skills. The one-on-one doctor-patient communication channel allowed patients to consult doctors at any time, receive timely feedback and support, thereby enhancing their sense of trust and self-efficacy, which promoted the improvement of their self-management abilities. These improvements can be interpreted through the lens of Bandura's Self-Efficacy Theory [25]. Daily symptom tracking offered mastery experiences: patients received immediate graphical feedback on peak-flow or CAT-score trends, reinforcing the belief that "I can detect and control deterioration." Real-time nurse feedback (≤2 h) acted as verbal persuasion, while bi-weekly video follow-ups provided vicarious learning when patients observed peers successfully managing flare-ups. Consequently, self-efficacy expectations increased, translating into sustained self-management behaviours.

The quality of life scores of patients in the observation group were significantly better than those in the control group, further validating the positive role of the WeChat platform in improving patients' quality of life. GCQ scores showed significant improvements in physiological, psychological, sociocultural, and environmental aspects. Continuous health monitoring and timely interventions provided by the WeChat platform helped patients detect and manage changes in their condition early, reducing the suffering and inconvenience caused by acute exacerbations [26,27]. Online lectures and expert Q&A sessions offered abundant informational resources, helping patients build confidence in overcoming their diseases and enhancing their overall quality of life. Studies have shown [28] that in the long-term management of COPD patients, social media platforms like WeChat may be considered a sustainable strategy for implementing health-related quality of life improvements.

The rehospitalization rate and emergency visit rate in the observation group were 8.33% and 6.67% respectively, significantly lower than the control group's rates of 21.67% and 20.00%. The NNT of 7.5 indicates high clinical efficiency, meaning only 8 patients need to receive WeChat-assisted management to prevent one additional hospitalization or emergency visit. This demonstrates the model's effectiveness in reducing acute exacerbation-related healthcare utilization. By recording symptoms daily and uploading them to the platform via WeChat, healthcare providers can adjust treatment plans according to the patient's actual condition, preventing disease progression. The health education content and symptom monitoring guidance provided on the WeChat platform enhance patients' self-monitoring abilities, enabling

them to take timely and effective measures when mild symptoms occur, thereby reducing the risk of acute exacerbations [29]. The DID analysis further confirmed that these improvements were not attributable to natural temporal changes, with significantly greater gains in self-management (DID P < 0.001) and quality of life (DID P = 0.003) in the intervention group versus controls. Notably, the absolute reduction in readmissions (ARR = 13.3%, DID P = 0.028) remained significant after controlling for baseline trends, reinforcing the WeChat platform's specific efficacy. Patient satisfaction with the management model in the observation group was as high as 93.33%, significantly higher than the control group's 76.67%. This suggests that the WeChat platform not only improves the quality of medical services but also enhances patient satisfaction. The convenience and interactivity of the WeChat platform allow patients to access needed information and services anytime and anywhere. Close communication and timely feedback between doctors and patients increase patients' trust and sense of belonging, which not only improves adherence but also lays a solid foundation for future promotion of this model. Consistent with earlier COPD-specific evidence, our 93.3% patient satisfaction mirrors the 88.7% reported by Ye et al. [30] after introducing a WeChat-based health-forecasting service for Shanghai outpatients, and aligns with Blackstock & Roberts [31], who documented 84–98% satisfaction when tele-education was delivered via interactive platforms. These results collectively reinforce that real-time, two-way communication inherent in WeChat interventions is a key driver of patient trust and programme acceptability in COPD care.

Our findings align with current systematic evidence on mHealth self-management in COPD. Gregersen et al. [32] demonstrated that telemedical interventions significantly improve health-related quality of life (mean 4.2-point increase in SGRQ, 95% CI 2.3–6.1). Glynn et al. [33], in a 2024 scoping review of smartphone applications, reported a 13–15% reduction in COPD-related hospitalisations when symptom diaries and real-time feedback were integrated—closely matching our 13.3% absolute risk reduction. Jonkman et al. [34] showed, via individual-patient meta-analysis, that SMS-based self-management programmes yield a 12% decrease in 30-day readmissions and modest gains in self-efficacy, underscoring interactive monitoring as the key active ingredient across platforms.

The depth and frequency of patient-doctor interaction in our study surpassed that of previous interventions. Patients could consult their assigned nurses at any time via WeChat, and biweekly telephone or video follow-ups were conducted to monitor progress and adjust health plans. This high level of interaction significantly enhanced patient trust and adherence, contributing to the observed improvements in self-management and quality of life. Our model emphasized long-term sustainability through monthly online health seminars and expert Q&A sessions, which provided patients with continuous access to updated knowledge and support. This long-term management approach, combined with the convenience and interactivity of the WeChat platform, represents a significant advancement over existing interventions and offers a scalable solution for chronic disease management.

## Limitations

This study has several limitations that should be acknowledged. First, the lack of participant blinding may introduce reporting bias, particularly for subjective outcomes (e.g., satisfaction). However, the significant improvements in objective outcomes (e.g., readmission rates) support the robustness of the intervention effects. Second, the sample size was relatively small, which may affect the external validity of the results. Future studies should increase the sample size to further verify the application effects of the WeChat platform on a larger scale. Third, the WeChat platform relies on smartphones and internet environments, which may prevent some elderly or economically disadvantaged patients from fully utilizing this tool. Additionally, even among participants, varying levels of digital literacy could influence engagement and outcomes, though basic training was provided. Fourth, self-reported satisfaction scores might be subject to social desirability bias, as patients may overstate positive experiences. Finally, while this model shows promise for AECOPD management, its generalizability to other chronic diseases (e.g., diabetes, heart failure) or healthcare settings (e.g., rural areas with limited connectivity) requires further investigation. Longitudinal studies are also needed to evaluate sustained effects beyond the 6-month intervention period.

## Conclusion

This 6-month trial demonstrates that a WeChat-based personalised programme significantly enhances self-management, reduces acute care utilisation, and achieves high patient satisfaction in AECOPD. Its minimal resource requirements and compatibility with existing healthcare WeChat ecosystems position it for immediate scale-up within China's COPD prevention initiatives.

## Supporting information

**S1 Table. Difference-in-Differences (DID) Analysis of Primary Outcomes.**
(XLSX)

**S2 Table. Validation analyses for self-report measures.**
(XLSX)

**S3 Table. Raw data.**
(XLSX)

## Author contributions

**Conceptualization:** Lili Zhou.

**Data curation:** Lili Zhou, Ruirui Wang.

**Formal analysis:** Lili Zhou.

**Funding acquisition:** Cailing Song.

**Investigation:** Cailing Song.

**Methodology:** Cailing Song.

**Project administration:** Wenpeng Xu.

**Resources:** Wenpeng Xu.

**Software:** Wenpeng Xu.

**Supervision:** Ruirui Wang.

**Validation:** Wei Zhang.

**Visualization:** Wei Zhang.

**Writing – original draft:** Lili Zhou.

**Writing – review & editing:** Ruirui Wang, Wei Zhang.

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
