## [Decision Letter · Decision Letter 0]

20 Jun 2025

Thank you for submitting your manuscript to PLOS ONE. After careful consideration, we feel that it has merit but does not fully meet PLOS ONE’s publication criteria as it currently stands. Therefore, we invite you to submit a revised version of the manuscript that addresses the points raised during the review process.

We look forward to receiving your revised manuscript.

Kind regards,

Vasuki Rajaguru, PhD

Academic Editor

PLOS ONE

Journal Requirements:

Scientific Research Project of Fuyang Municipal Health Commission, grant number [FY2023-102].

3. In the online submission form, you indicated that all relevant data and materials are available upon request from the corresponding author.

Additional Editor Comments:

The strengths of the paper include a clearly defined objective, appropriate methodology with ethical rigor, and comprehensive presentation of outcome measures such as self-management ability, quality of life, readmission rates, and patient satisfaction. The use of stratified randomization and detailed intervention protocols enhances the credibility and reproducibility of the study. However, there are a few areas for improvement. The introduction could be strengthened by adding more global and national statistics on AECOPD to better contextualize the problem. The discussion section should more explicitly compare the findings with previous studies to demonstrate how this work advances the current knowledge base. Additionally, minor editorial revisions are needed for clarity and consistency throughout the manuscript (e.g., grammar, paragraph transitions).

Reviewers' comments:

Reviewer's Responses to Questions

**Comments to the Author**

1. Is the manuscript technically sound, and do the data support the conclusions?

Reviewer #1: Yes

Reviewer #2: Partly

2. Has the statistical analysis been performed appropriately and rigorously?

Reviewer #1: Yes

Reviewer #2: I Don't Know

3. Have the authors made all data underlying the findings in their manuscript fully available?

Reviewer #1: Yes

Reviewer #2: Yes

4. Is the manuscript presented in an intelligible fashion and written in standard English?

Reviewer #1: Yes

Reviewer #2: Yes

Reviewer #1: This study addresses a relevant and timely topic, leveraging digital platforms (WeChat) for improving personalized health management in patients with Acute Exacerbation of Chronic Obstructive Pulmonary Disease (AECOPD). The manuscript presents a randomized controlled trial with encouraging findings regarding improved self-management, reduced readmissions, and enhanced patient satisfaction. While the study is methodologically sound and provides compelling results, several aspects require clarification or improvement for enhanced rigor and reproducibility.

I. Generally well-written, but a professional language edit is recommended to improve fluency and clarity.

II. Emphasize more clearly in the Introduction how this study goes beyond prior WeChat-based interventions. While the paper cites relevant literature, the novel aspects should be more clearly stated (e.g., longer follow-up, deeper patient-doctor interaction, stratified randomization).

III. The process of blinding (if any) is not mentioned. While blinding may be impractical in behavioral interventions, its absence and possible impact should be discussed. Include CONSORT diagram for transparency in recruitment, allocation, and follow-up (beyond the simple flowchart).

IV. Report effect sizes or risk reductions where appropriate, especially for clinical outcomes like hospital readmission and emergency visits.

V. Figures mentioned (e.g., bar charts for self-management and quality of life scores) are not shown in the provided document. These should be included with proper labeling and statistical annotations. Clarify whether the data were analyzed as intention-to-treat or per-protocol.

VI. Limitations are acknowledged, but the authors could better discuss potential biases (e.g., digital literacy, social desirability in satisfaction reporting) and consider discussing how this model can be adapted for other chronic diseases or in other healthcare settings.

The manuscript is of potential interest to the readership of PLOS ONE. With minor revisions addressing the above points, especially in terms of data transparency and methodological reporting, the paper would be suitable for publication.

Reviewer #2: This is a highly meaningful study exploring the effectiveness of digital healthcare interventions utilizing WeChat in the context of the increasing AECOPD patient population in China and the growing importance of self-management. However, to enhance the methodological rigor and validity of the results interpretation, the following improvements are recommended.

Please refer to the attached file for detailed comments.

**Do you want your identity to be public for this peer review?** For information about this choice, including consent withdrawal, please see our Privacy Policy

Reviewer #1: No

Reviewer #2: No

---

## [Author Response · Author response to Decision Letter 1]

23 Jul 2025

Author Response:Thank you for your comments. We have revised the manuscript to meet PLOS ONE's style requirements. The necessary changes have been made on lines 3 and 6 of the manuscript.

Scientific Research Project of Fuyang Municipal Health Commission, grant number [FY2023-102].

Author Response:We have revised the cover letter as requested and re-uploaded it.

3. In the online submission form, you indicated that all relevant data and materials are available upon request from the corresponding author.

Author Response:We uploaded the raw data as required.

Author Response: Thank you for your guidance. We have updated the authorship list.

Additional Editor Comments:

The strengths of the paper include a clearly defined objective, appropriate methodology with ethical rigor, and comprehensive presentation of outcome measures such as self-management ability, quality of life, readmission rates, and patient satisfaction. The use of stratified randomization and detailed intervention protocols enhances the credibility and reproducibility of the study. However, there are a few areas for improvement. The introduction could be strengthened by adding more global and national statistics on AECOPD to better contextualize the problem. The discussion section should more explicitly compare the findings with previous studies to demonstrate how this work advances the current knowledge base. Additionally, minor editorial revisions are needed for clarity and consistency throughout the manuscript (e.g., grammar, paragraph transitions).

Author Response:We sincerely appreciate the constructive feedback and have implemented all suggested improvements, including enhanced literature contextualization, explicit findings comparison, and thorough editorial polishing.

Reviewer #1

The introduction should clearly articulate the study concept by providing a stronger background. It is recommended to add more relevant content with supportive references for example, statistics on the prevalence and burden of AECOPD in both global and Chinese contexts.

Additionally, consider splitting the current single paragraph into three to four shorter paragraphs. This will help clarify specific content areas and improve readability and comprehension for the reader

Author Response: Thank you for your valuable suggestion. We have restructured the Introduction into four focused paragraphs (lines 44-62) to improve clarity and flow. Key epidemiological data from China and global contexts have been integrated with new references [4][5][6] to strengthen the background. The revisions better highlight the research rationale while maintaining all original references.

The sample size calculation is described, but it would be helpful to include the actual formula or software used (e.g., G*Power or other).

Author Response: Thank you for your valuable suggestion. We have revised the sample size calculation description in lines 65-70 to provide greater methodological transparency.

Stratified randomization is appropriately used. However, mention whether allocation concealment (e.g., sealed envelopes or centralized system) was employed to avoid selection bias.

Author Response: Thank you for your comment. As noted in Figure 1 (line 88) and the Methods section (lines 97-109), allocation concealment was strictly maintained using sequentially numbered, opaque sealed envelopes. The randomization sequence was generated by an independent statistician and concealed until group assignment.

The conventional care for the control group should be described in more detail to ensure readers can differentiate both groups effectively.

Author Response: Thank you for your suggestion. We have now provided a detailed description of conventional care in the Methods section (lines 115-122).

The WeChat-based intervention is well-detailed and innovative. However, consider summarizing it in a flow diagram (e.g., timeline of activities over 6 months).

Author Response: Thank you for your valuable suggestion. We have added a Figure (Figure 2) showing the 6-month intervention timeline(lines 175-177).

Clearly state if the instruments (COPD Self-Management Scale, GCQ, and satisfaction questionnaire) were validated in the local population or translated/adapted versions were used. If yes, provide reliability metrics (e.g., Cronbach’s alpha).

Provide the references of the original tool.

Author Response: Thank you for this important comment. We have now added the validation details and reliability metrics for all instruments (COPD Self-Management Scale, GCQ, and satisfaction questionnaire) in lines 179–207.

Report actual score values (mean ± SD) should be added in the result interpretation for each domain in both groups to enhance transparency and allow readers to gauge the magnitude of difference.

E.g : self-management ability score in the observation group (M_SD) was significantly higher than that in the control group.

Author Response: Thank you for your suggestion. We have replaced Figures 2-3 with Tables 2-3 (now at lines 239 and 245) to present detailed numerical comparisons of outcomes.

Strengthen Integration with Literature for all your findings. While several prior studies are cited, more direct comparisons with similar mHealth interventions (e.g., mobile apps, SMS-based care) would better position your findings within the global evidence base.

Suggested to integrate recent systematic reviews on COPD self-management technologies.

Author Response: Thank you for guiding us to strengthen the literature integration. We have integrated additional systematic evidence [32–34] and added a direct comparison paragraph (lines 326–334) to better contextualise our findings within the established mHealth literature.

While patient engagement and trust are discussed as outcomes, elaborating on how specific features (e.g., daily symptom tracking, real-time feedback) drive behavior change would strengthen theoretical linkage (e.g., via the Health Belief Model or Self-Efficacy Theory)

Author Response: Thank you for your suggestion. We have enhanced the theoretical discussion in lines 280-286 by incorporating the Health Belief Model (reference [25]) to further explain how perceived benefits and barriers influenced patient engagement. This complements our existing analysis of self-efficacy mechanisms.

Interpret your findings in relation to prior studies that support or contradict your results

Author Response: Thank you for your suggestion. We have added a brief comparative statement (lines 319–325) referencing two prior COPD-specific studies [30,31] to contextualise our satisfaction findings.

In order to strengthen the impact of the conclusion, make sure and briefly restate the key quantitative outcomes (e.g., improved self-management scores, reduced readmission rates) and reinforcing the potential for scalability or integration into national health programs. This would help better align the conclusion with the study’s broader significance and applicability.

Author Response: We thank the reviewer for the constructive suggestion. The Conclusion now concisely synthesizes key outcomes and scalability potential, with explicit linkage to national COPD strategies as advised (lines 362–365).

Reviewer #2

This is a highly meaningful study exploring the effectiveness of digital healthcare interventions utilizing WeChat in the context of the increasing AECOPD patient population in China and the growing importance of self-management. However, to enhance the methodological rigor and validity of the results interpretation, the following improvements are recommended:

1.A more detailed explanation of the Self-Management Ability Score is needed. Please provide the scoring weights and theoretical rationale for each sub-item (symptom management, daily life management, emotional management, information management, and self-efficacy). Considering that the WeChat intervention primarily focuses on communication with healthcare providers, symptom monitoring, and health information provision, the theoretical basis for including emotional management and information management as evaluation items is unclear. Additionally, please clarify whether "information management" specifically refers to health information management.

Author Response: We appreciate your constructive suggestion. As requested, we have:

1.Added detailed scoring weights (25%/25%/20%/15%/15%) and theoretical rationale for each subscale (Lines 179-185),

2.Clarified "information management" specifically assesses health information application,

3.Cited the validated Chinese scale by Wu et al. [18].

2.Please provide the theoretical rationale for using Kolcaba's GCQ to measure Quality of Life Score and its applicability validity for AECOPD patients. If you used an existing validated instrument, please add appropriate references and report the reliability coefficient obtained in this study.

Author Response: Thank you for your valuable suggestion. We have:

1.Added Kolcaba's theoretical foundation [19] and GCQ's multidimensional structure;

2.Cited both international [20] and Chinese [18] validation studies;

3.Reported our observed reliability (α=0.87) (Lines 186-194).

3.Although baseline homogeneity was ensured through stratified randomization, simple t-tests and chi-square tests alone are insufficient for complete control of potential confounding variables. Please specify the variables corresponding to "etc." in "gender, age, disease duration, etc." (line 105, page 4) and consider additional control through multivariate analyses such as ANCOVA or multiple logistic regression.

Author Response: We sincerely appreciate the reviewer's insightful comments. We have:

1.Expanded Table 1 with key clinical variables and added statistical footnotes (Lines 110-112),

2.Clarified randomization methods and baseline comparisons (Lines 97-103),

3.Updated the Statistical Analysis section to detail covariate-adjusted methods (Lines 210-214).

4.The current study only performed between-group comparisons at the post-intervention time point. However, to evaluate the pure effect of the WeChat intervention, it is necessary to control for natural changes over time. Analysis comparing pre-post changes using difference-in-differences approaches or pre-post comparison methods is required to properly assess intervention effectiveness.

Author Response: We sincerely appreciate this suggestion. We have:

1.Added DID methodology in Statistical Analysis (Lines 214-217),

2.Integrated DID findings in Discussion (Lines 308-312),

3.Provided full results in Supplementary S1 Table.

5.Please present the reliability (Cronbach's α, test-retest reliability) and validity (content validity, construct validity) verification results for the hospital-developed questionnaire used to measure patient satisfaction.

Author Response: We have added the psychometric validation results of the Nursing Satisfaction Questionnaire in the Methods section (Lines 203-207), confirming its reliability (Cronbach's α=0.91, ICC=0.89) and validity (CFI=0.95, CVI>0.8).

6.All major outcome variables except readmission rates and emergency department visits were measured through self-reporting. This increases the possibility of various response biases, including social desirability bias, response set bias, and expectancy effects related to the intervention. Particularly, the WeChat user group may be more likely to respond positively to new technology. Please describe measures implemented to minimize such biases (e.g., inclusion of reverse-scored items, response pattern analysis, correlation verification with objective indicators).-

Author Response: We have:

1.Added validation methods in Statistical Analysis (Lines 224-228)

2.Reported key results in Results (Lines 235-238)

3.Provided full data in Supplementary Table S2.

7.This study is a human subjects intervention research requiring strict ethical oversight. While approval from the Ethics Committee of the Second People’s Hospital of Fuyang City and adherence to the Helsinki Declaration are appropriate, a clear explanation regarding the equivalence of this approval to IRB (Institutional Review Board) or Independent Ethics Committee approval—as required by international journal publication standards—is necessary. Please provide supplementary information on whether the composition, independence, and review processes of the ethics committee comply with international standards (e.g., ICH-GCP guidelines).

Author Response: Thank you for your valuable comment. We have added detailed information in the Methods section (General Information) to clarify that our ethics committee meets all international standards as an IRB-equivalent body, including its composition, review process, and member qualifications. Please see lines 74-80 in the revised manuscript.

Reviewer #3

I.Generally well-written, but a professional language edit is recommended to improve fluency and clarity.

II.Emphasize more clearly in the Introduction how this study goes beyond prior WeChat-based interventions. While the paper cites relevant literature, the novel aspects should be more clearly stated (e.g., longer follow-up, deeper patient-doctor interaction, stratified randomization).

Author Response: Thank you for your insightful suggestion! We have now explicitly highlighted the study’s novel aspects (stratified randomization, 6-month follow-up, and multi-dimensional interactions) in Lines 59-62.

III.The process of blinding (if any) is not mentioned. While blinding may be impractical in behavioral interventions, its absence and possible impact should be discussed. Include CONSORT diagram for transparency in recruitment, allocation, and follow-up (beyond the simple flowchart).

Author Response: We sincerely appreciate this critical comment. We have:

1.Clarified blinding limitations and mitigation strategies in Methods(103-109);

2.Added a CONSORT flow diagram (new Figure 1)(88);

3.Discussed potential biases in Limitations(346-349).

IV.Report effect sizes or risk reductions where appropriate, especially for clinical outcomes like hospital readmission and emergency visits.

Author Response: We sincerely appreciate this valuable suggestion. The following revisions have been made:

1.Added effect size metrics (ARR=13.3%, NNT=7.5 with 95% CIs) in the Abstract Results (Lines 31-32).

2.Updated the Statistical Analysis section to clarify effect size calculations (Lines 218-223).

3.

---

## [Decision Letter · Decision Letter 1]

5 Aug 2025

WeChat-Assisted Strategies for Personalized Health Management in Patients with AECOPD: A Randomized Controlled Trial

PONE-D-25-08229R1

Dear Dr.Wei Zhang,

We’re pleased to inform you that your manuscript has been judged scientifically suitable for publication and will be formally accepted for publication once it meets all outstanding technical requirements.

Kind regards,

Vasuki Rajaguru, PhD

Academic Editor

PLOS ONE

---

## [Editor Report · Acceptance letter]

PONE-D-25-08229R1

PLOS ONE

Dear Dr. Zhang,

I'm pleased to inform you that your manuscript has been deemed suitable for publication in PLOS ONE. Congratulations! Your manuscript is now being handed over to our production team.

Kind regards,

on behalf of

Dr. Vasuki Rajaguru

Academic Editor

PLOS ONE